# Is Complete Excision Always Enough? A Quality of Sexual Life Assessment in Patients with Deep Endometriosis

**DOI:** 10.3390/medicina60091534

**Published:** 2024-09-20

**Authors:** Raluca Gabriela Enciu, Octavian Enciu, Dragoș Eugen Georgescu, Adrian Tulin, Adrian Miron

**Affiliations:** 1Clinical Hospital of Obstetrics and Gynecology “Prof. Dr. Panait Sârbu”, 060251 Bucharest, Romania; ralucagabrielasuba@gmail.com; 2Medicover Endometriosis Center, 013982 Bucharest, Romania; 3Discipline of General Surgery, Faculty of Medicine, “Carol Davila” University of Medicine and Pharmacy, 050474 Bucharest, Romania; dragos-eugen.georgescu@umfcd.ro (D.E.G.); adrian.tulin@umfcd.ro (A.T.); adrian.miron@umfcd.ro (A.M.)

**Keywords:** endometriosis, EHP-30, dyspareunia, QoSL

## Abstract

*Background and Objectives*: The aim of this study was to find the factors associated with the severe impairment of QoSL and the factors associated with a better score in QoSL, as well as the evaluation of pain symptoms and QoSL after the complete and incomplete excision of rectovaginal nodules. *Materials and methods*: The present prospective study was conducted in a single tertiary center for endometriosis where 116 patients underwent laparoscopic surgery for deep endometriosis during a 3-year period. The goal of the intervention was to excise all endometriotic implants while conserving the rectum. Intraoperative findings were recorded after the intervention, and the patients were classified according to the ENZIAN classification and rASRM scores. QoSL was assessed using the EHP-30 Module C (QoSL Score). *Results*: When comparing the mean scores before and 2 years after the surgery, a highly significant improvement was found for QoSL and dysmenorrhea (*p* < 0.0001). The complete excision of rectovaginal nodules led to a significantly better QoSL and lower dyspareunia (*p* < 0.0001) than incomplete resection (*p* < 0.02). *Conclusions*: This prospective study proves that the complete laparoscopic excision of all endometriotic implants improved the QoSL and decreased the pain score of dyspareunia. Incomplete rectovaginal nodule excision was correlated with a poorer QoSL and a lower improvement of dysmenorrhea, dyspareunia, and chronic pelvic pain scores than complete excision.

## 1. Introduction

Endometriosis is a gynecologic disease defined by the presence of endometrial glands and stroma outside the uterus [1]. Due to the difference in site, pathogenesis, and hormonal responsiveness, peritoneal, ovarian, and rectovaginal endometriosis can be considered as three distinct entities [2]. Endometriotic lesions that infiltrate deeper than 5 mm under the peritoneum are considered deep endometriosis [3].

The current literature describes “deep infiltrating endometriosis” as extensive adhesions in the cul-de-sac, obliterating its lower portion and affixing the cervix or the lower part of the uterus and the rectum together [4]. Deep infiltrating endometriosis commonly affects the uterosacral ligaments, the rectum, the vagina, the rectovaginal septum, and the Pouch of Douglas [5,6,7]. Using the ENZIAN classification, the assessment of endometriotic tissue infiltration into the rectovaginal septum, the uterosacral ligaments, the vagina, and the intestine wall is more accurate [8].

As a result of destructive alterations at different sites produced by endometriosis, this disease is frequently associated with complex symptoms like dysmenorrhea, deep dyspareunia, chronic pelvic pain, and dyschezia. Several studies have concluded that the severity of the pain does not correlate with the disease stage [9,10,11]. On the other hand, it has been observed that the severity of the pain correlates with the depth of infiltration [1,12].

The etiopathogenesis of the pain related to endometriosis is multifactorial: from local inflammatory mediators such as prostaglandins, histamine, kinins, and interleukins that eventually lead to pelvic adhesions and spasticity of the pelvic floor muscles, to neurogenic inflammation due to the ingrowth of nerve fibers into endometriotic lesions and central sensitization translated as an exaggerated response to a normal stimulus [13,14,15,16].

Dyspareunia is defined as the genital pain experienced just before, during, or after sexual intercourse, and is classified as superficial/entry dyspareunia, resulting from conditions affecting the labia or vestibule, and deep dyspareunia, more frequently associated with endometriosis, pelvic adhesions, and pelvic inflammatory disease [17].

Although Vercellini and Chapron observed a strong association between dyspareunia and deep posterior cul-de-sac lesions and several authors proved that uterosacral ligament involvement is correlated with deep dyspareunia, in light of all the above, it may be considered naïve to assume that only the localization and characteristics of the endometriotic implants explain the type of symptoms [11,18,19,20].

The treatment of endometriosis should focus on symptom reduction and quality of life enhancement. Even though controversy reigns over the management of deep infiltrating endometriosis, there is a general consensus that all visible endometriotic lesions should be surgically removed completely while taking great care to minimize ovarian damage [21,22,23].

Estimating the quality of life using only visual analog scales is very simple but inexact, thus highlighting the need for more complex tools of measurement. Of the many disease-specific tested and validated questionnaires, EHP-30 seems to be the best suited.

## 2. Materials and Methods

The present work was conducted in a single tertiary center for endometriosis in Bucharest, Romania, at the public teaching hospital “Prof. Dr. Panait Sîrbu” Clinical Hospital. The prospective experimental study design was created before enrolling patients and is considered suitable for the purpose of statistical analysis. The study has obtained the approval of the hospital ethics committee.

The enrolled patients had to meet the following inclusion criteria: heterosexual patients aged between 18 and 39 years with severe deep endometriosis (stage III-IV rASRM), dyspareunia lasting for at least 1 year, and have not received GnRH analogs during the last three months. The patients who agreed to take part in the study received and signed the informed consent for surgery and completed the EHP-30 quality of life questionnaire both prior to the surgery and at the follow-up. All studied patients had histologically confirmed endometriosis.

The following were considered exclusion criteria: psychiatric disorders, use of drugs that could affect cognition, pain of a different origin than endometriosis, virgo intacta or no sexual intercourse during the last year, and a history of sexual assault.

The consent rate was 97.7%, with only 3 patients refusing to participate in this research. Twelve patients (9.37%) were lost to follow-up at 2 years, and the response rate was 90.6%.

Demographics and specific data (menarche, sex life, abortion, pregnancy, history of infertility and gynecologic surgeries, and intraoperative data) were recorded using Microsoft Office Excel™ (v16.0, Microsoft, Redmond, WA, USA).

The patients completed the quality of life questionnaire the day before the surgery, alone in the hospital room and without intervention from the medical staff. Using the same questionnaire sent by email or postal service, the patients were re-evaluated 1 year and 2 years after the surgery. The results at 2 years were considered to be more relevant given the time needed to heal both physically and emotionally and to allow for centralization mechanisms to fade away; thus, this paper focuses on comparing the quality of sexual life (QoSL) prior to the surgery and 2 years after the surgery. Data were pooled using Microsoft Office Excel™.

A complete database was developed in Microsoft Excel^®^, and add-ins like WinSTAT (R. Fitch Software, Cambridge, MA, USA) and XLSTAT (Lumivero, Denver, CO, USA) were used to perform Pearson’s (r) and Spearman’s (rho) tests to certify correlations. For *p* values under 0.05, the results were considered statistically significant, while for *p* values less than 0.001, the results were considered highly significant. A *t*-test was used for determining differences between the means of groups for variables with a normal distribution.

Surgical Intervention—standard of practice

Antibiotic and deep vein thrombosis prophylaxis were carried out according to international guidelines. Mechanical bowel preparation was carried out in all patients with Fortrans^®^.

The surgical intervention was performed under general anesthesia with the patient in lithotomy position. Having no ERAS protocols implemented for gynecological surgery, all patients received general anesthesia using Propofol for induction, opioids (Fentanyl) for pain management, and Esmeron and an inhalational anesthetic (Sevoflurane) for curarization. None of the patients received epidural anesthesia. The uterine manipulator provided with a probe for tubal patency testing with methylene blue dye was placed in position before commencing the intervention. Betadine rectal washout was performed prior to entering the peritoneal cavity.

The goal of the intervention was to excise all endometriotic implants while conserving the rectum, even if sometimes the dissection was very close to the rectum and shaving of the serosa or outer muscular layers was required.

Briefly, the laparoscopic surgical steps were the following: inspection of the pelvic cavity; adhesion management and freeing of the ovaries (cyst effraction, content drainage, and ovariopexy, when necessary); freeing the sigmoid colon from adhesions and frequently from the natural coalescence; ureterolysis starting from the iliac part to the paracervical region; dissection and shaving of nodules from the uterosacral ligaments (USL), torus, and vagina; opening of the pararectal spaces and release of the lateral parts of the rectum; and then removing the rectum from the posterior aspects of the uterus and vagina and shaving the nodules in this area using monopolar cautery. In cases with vaginal wall involvement, limited vaginal resection was performed. The final surgical steps included an ovarian cystectomy using the stripping technique and the tubal patency test. When a rectal injury was suspected, a rectal instillation test with methylene blue dye was performed.

Intraoperative findings were recorded after each intervention, and the patients were classified according to the ENZIAN classification and rASRM scores [8,24,25].

## 3. Results

Of all the suitable patients for the study, twenty-seven were excluded: four patients took GnRH analogs before surgery, two patients were diagnosed with depression, two patients were virgo intacta, four patients did not have sexual intercourse over the past year, three patients refused to enroll in the study, and twelve patients refused to complete the follow-up questionnaire 2 years after the surgery.

After all the inclusion and exclusion criteria were applied, a group of 116 patients was constituted. The mean age was 30.7 ± 4.4 (range 22–39 years), the body mass index was 20.5 ± 3.4 kg/m^2^ (range 15–30), 37 patients consumed tobacco products (31.89%), and 79 did not (68.10%). Regarding residency, most patients (89, 83.62%) lived in an urban setting, and more than half (59, 50.86%) had higher education. Regarding employment status, 83.19% were employed, 10.34% were homemakers, and 4.31% were students. Twenty-five percent had received at least one previous surgical intervention for endometriosis in another center. All of the patient’s descriptive characteristics are summarized in Table 1.

Twenty-three (19.80%) patients became pregnant before receiving surgery in the current study; of these, eight patients had a vaginal delivery and the rest delivered with a C-section. In the current study, seventy-nine patients (68.10%) expressed their intention to become pregnant and only forty-two (53.20%) became pregnant—twenty-nine spontaneously, five using artificial insemination, and eight by in vitro fertilization. The rest of the patients underwent treatment with oral estro-progestins or progestin-only contraceptives after surgery.

Surgical findings are highlighted in Table 2 and surgical procedures are indicated in Table 3. Endometrial peritoneal implants were found in all cases, most frequently being located beneath the ovarian cysts, in the ovarian fossa. We performed 69 right ovarian cystectomies, 92 left ovarian cystectomies, and 86 complete or near complete adhesiolysis in the Pouch of Douglas. Of all eighty-two patients with rectovaginal nodules, we achieved complete excision in seventy-three cases and partial excision in nine cases. In certain circumstances, like bleeding and risk of rectal and hypogastric plexus injury, rectovaginal nodules were only partially excised. Eighty-eight patients required a uterosacral ligament resection, eleven patients required a vaginal wall resection, and four patients needed a bladder nodule excision.


**Evaluation of dyspareunia and QoSL before laparoscopy**


The distribution of patients by the intensity of dyspareunia using a visual analog score (Figure 1) with and without anti-inflammatory medication shows that 16.37% of patients perceived no pain (0-VAS), followed by 13.79% of patients that perceived moderate pain (5-VAS), and 5.17% of patients that reported severe or unbearable pain (10-VAS).

Figure 2 emphasizes the linear correlation with high statistical significance between dyspareunia and the medium score of Sexual QoSL (EHP-30 Module C) (*p* < 0.0001). However, the pain scale is insufficient for the assessment of surgical efficiency because the information is very limited. Illness-specific validated questionnaires bring far more detailed evidence [26].

After analyzing the collected data, several factors appeared to be associated with the quality of sexual life, and their correlation was tested. These factors are summarized in Table 4.

Factors associated with severe impairment of quality of sexual life:Personal history of surgical interventions for endometriosis (Pearson correlation coefficient r = 0.1895, *p* = 0.04);Completely obliterated Pouch of Douglas (Pearson correlation coefficient r = 0.2860, *p* = 0.001);More severe score in the Pain domain of the EHP-30 core (Pearson correlation coefficient r = 0.2969, *p* = 0.001).

Factors associated with a better score of quality of sexual life:Unobliterated Pouch of Douglas (Pearson correlation coefficient r = 0.2023, *p* = 0.02);Lesion-free uterosacral ligaments (Pearson correlation coefficient r = 0.2321, *p* = 0.01).

Preoperative versus 2-years-after-surgery results comparison regarding dyspareunia and QoSL

The statistical comparison of the mean scores revealed that pain intensity during intercourse is correlated, with a very high statistical significance (*p* < 0.0001), with QoSL before and 2 years after the surgery, meaning that 2 years after the surgery, the pain intensity during intercourse is much lower and QoSL shows a great improvement (Table 5).

The difference between the quantified values before and 2 years after surgery is highly significant (*p* < 0.001) for all pain types (dyspareunia, dysmenorrhea, and chronic pelvic pain).

**Complete vs. partial resection of rectovaginal nodule** (Table 6)

The comparison of the mean scores demonstrated a highly significant improvement (*p* = 0.02) of QoSL 2 years after the surgery for patients with complete resection of rectovaginal nodules vs. patients with partial resection. Partial resection of the rectovaginal nodule is poorly correlated (Spearman correlation coefficient rho = 0.1979) with a low QoSL, and this correlation is statistically significant (*p* = 0.03).

After comparing the mean scores for dyspareunia and dysmenorrhea 2 years after the surgery, a highly significant difference was found (*p* < 0.0001): patients who had undergone the complete excision of the rectovaginal nodule reported a lower score for pain.

After comparing the mean scores for chronic pelvic pain 2 years after the surgery, a significant difference was found (*p* = 0.02): patients who had undergone the complete excision of the rectovaginal nodule reported a lower score for pain.

## 4. Discussion

At the present time, the management of endometriosis-related pain focuses on four aspects: complete surgical excision of lesions, ovarian suppression of estradiol secretion, anti-inflammatory treatment, and efforts to treat pain directly by administration of analgesic drugs.

These approaches are effective in most of the patients with endometriosis, but in particular cases, the failure of these efforts is not understood.

There is a variable response to different treatments, probably related to the complex genetic and environmental factors that influence the process in each individual woman, which makes the study of pain and its origins more challenging.

The interventions were carried out systematically by a single team consisting of two gynecologists and two surgeons with advanced laparoscopic skills who had gotten past the learning curve for endometriosis resections. Regarding complications, the surgical difficulty of the cases that have had previous endometriosis surgery (29 patients, 25%) was not increased, mainly because they did not have previous extensive pelvic dissection. Of the eighty-two patients that required resection of the rectovaginal nodule, two had rectal perforations that required a rectal laparoscopic suture (PGA 2.0). There was no rectal or rectovaginal fistula and no rectal stricture in the long term. Two patients had postoperative bleeding in the first 12 h after the surgery, which was managed conservatively and did not require reintervention. A urinary catheter was left in place for 6 weeks for two patients needing extensive dissection of the uterosacral ligaments, with no difficulties afterward and no atonic bladder being diagnosed in the long term. One notable case had uroperitoneum diagnosed 4 weeks after the surgery, which was managed surgically in another center; it was a difficult dissection of the vesicouterine space with a late onset vesicoperitoneal fistula. Overall, the complication rate was 6.03%. The mean duration of the surgery was 128.45 min (range 30–220 min). No conversion to open surgery was necessary.

In this study, we proved that laparoscopic surgery with the complete excision of all endometriotic implants improved the quality of life from the sexual point of view and decreased the pain score of the most frequent symptoms produced by endometriosis. Despite all of the above, dysmenorrhea at the 2-year follow-up was zero only in three out of one hundred and sixteen patients (2.58%) and dyspareunia was reported as zero in thirty of one hundred and sixteen patients (25.86%). From the chronic pelvic pain perspective, more than half of the patients in the research group were pain-free (61, 52.58%).

Incomplete rectovaginal nodule excision is correlated with a poorer QoSL and a lower improvement of dysmenorrhea, dyspareunia, and chronic pelvic pain scores than complete excision.

The quality of life was studied 2 years after the surgery to allow enough time for physical and emotional healing because surgery can have a powerful emotional impact [27]. Pain during intercourse for a prolonged period can trigger centralization mechanisms that can act autonomously even in the absence of peripheral input from endometriosis lesions [28]. This fact can explain the persistence of pain in different degrees even after the complete excision of lesions.

Dyspareunia contributes as a major factor to quality of life impairment [29]. Each of the very few relevant studies concerning this issue used a different tool to test the quality of life: Ferrero used the Global Sexual Satisfaction Index and found that the score was significantly lower in women with deep endometriosis infiltrating the USL than in controls; Fritzer used the Female Sexual Function Index (FSI) and the Female Sexual Distress Scale (FSDS) and proved that FSI scores improved after 10 months from the operation in patients with peritoneal and deep endometriosis, but the FSDS score improved only in patients with deep endometriosis; Leona K. Shum used the EHP-30 and revealed that severe dyspareunia was associated with a worse EHP-30 Sexual QoL score, independent of superficial dyspareunia [30,31,32]. Regarding the inclusion criteria for the patients in this study, heterosexual patients were included, and virgo intacta patients and patients with no sexual intercourse in the last year were excluded in order to assess the degree of dyspareunia in a standardized manner. Dyspareunia among homosexual women is known to include a unique relational and social context and to be associated with advantages in pain management [33].

In this study, a personal history of surgical intervention for endometriosis, Pouch of Douglas obliteration, and a more severe score in the Pain domain of the EHP-30 core were associated with a serious impairment of QoSL. On the other hand, lesion-free uterosacral ligaments and an unobliterated Pouch of Douglas were associated with a better QoSL score.

It is well acknowledged that the pain mechanism in endometriosis is multifactorial and the ideal treatment is yet to be found, but we must highlight the important role of the surgeon. The aim of the surgical intervention must be accurate complete resection. The surgical treatment of endometriosis is challenging even for the most experienced surgeons, and the complexity of diagnosis and management needed for the complete resection of endometrial implants echo those of resections for cancer [34]. The current trend is to refer patients with deep endometriosis to specialized centers [35]. In repeated surgical interventions, incomplete endometriotic lesion excision may produce damage to nerve fibers and create the conditions for the development of neuropathic pain that could explain the poor results in those patients and the absence of pain relief.

This study’s limitations are a relatively low cohort of patients and the lack of detailed characteristics in the EHP-30 modular sexual domain. On the other hand, the fact that all interventions were carried out by the same team using the same operative strategy accounts for the absence of intraoperative variability. The follow-up after 2 years is significant: the patients had time to overcome the emotional postinterventional shock, to recover from the pain related to the surgery, and to create an overall image of the symptoms during intercourse and the impact on QoSL [36].

## 5. Conclusions

The present study indicates that patients with severe endometriosis (stages III-IV rASRM) who benefited from the laparoscopic excision of all the endometriotic implants had an overall improvement of the QoSL and a reduction in pain symptoms.

The reduction in endometriosis-related pain symptoms noticed in the majority of patients 2 years after the surgery led, most likely, to an improvement of QoSL.

Endometriosis remains an enigmatic disease, and even though it has been recognized as a specific pathological condition for over 100 years, we are still only at the very beginning of understanding its most remarkable and hurtful symptom, pain.

## Figures and Tables

**Figure 1 medicina-60-01534-f001:**
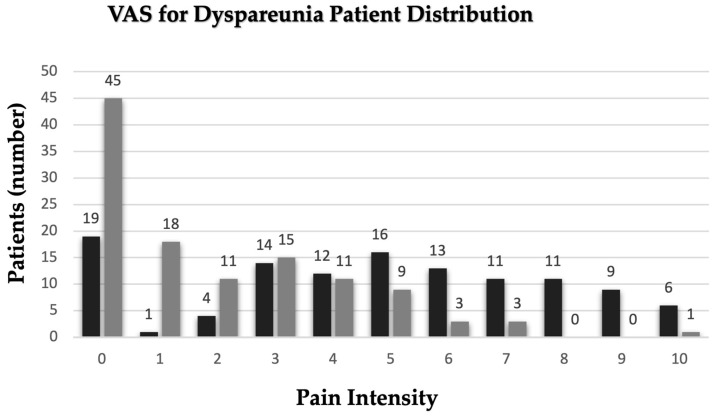
VAS for dyspareunia—patient distribution. Black columns—without medication; grey columns—with medication.

**Figure 2 medicina-60-01534-f002:**
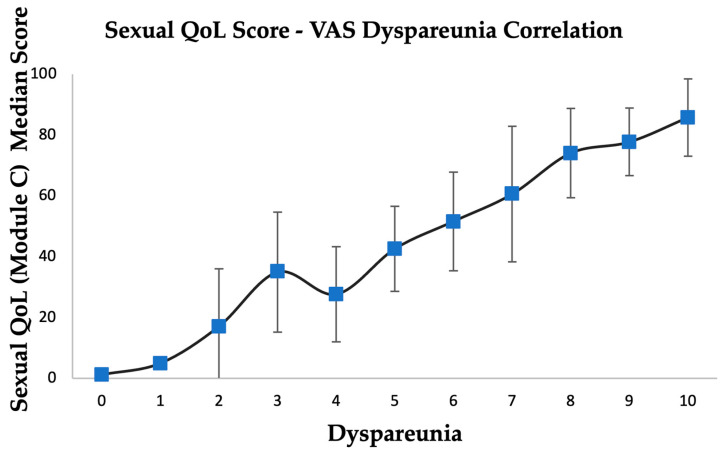
Sexual QoL score—VAS dyspareunia correlation.

**Table 1 medicina-60-01534-t001:** Descriptive characteristics of the studied patients.

		Mean ± SD/%
**Age (years)**		30.7 ± 4.4
**BMI**		20.5 ± 3.4
**Tobacco Consumption** **(cigarette smoking)**	Yes	37 (31.89%)
	No	79 (68.10%)
**Residence**	Urban	97 (83.62%)
	Rural	19 (16.37%)
**Education**	Primary education	3 (2.58%)
	Secondary education	54 (46.55%)
	Tertiary education (University)	59 (50.86%)
**Employment**	Employed	99 (83.19%)
	Homemaker	12 (10.34%)
	Student	5 (4.31%)
**Marital Status**	Married	70 (60.34%)
	In a relationship	43 (37.91%)
	Divorced	2 (1.72%)
**History of Prior Surgery for Endometriosis**	Yes	29 (25.00%)
	No	87 (75.00%)
**Pregnancy obtained before surgery**	Yes	23 (19.80%)
	No	93 (80.20%)
**Pregnancy obtained after surgery**	Spontaneous	29 (25%)
	Artificial insemination	5 (4.30%)
	In vitro fertilization	8 (6.90%)

**Table 2 medicina-60-01534-t002:** Surgical findings.

		Mean ± SD/%
**Right ovarian cyst**	Dimension (cm)	4.78 ± 1.83
**Left ovarian cyst**	Dimension (cm)	4.94 ± 1.59
**Pouch of Douglas obliteration**	Complete	55 (47.41%)
	Incomplete	31 (26.72%)
	Absent	30 (25.86%)
**Uterosacral ligament infiltration**	Bilateral	22 (1.96%)
	Unilateral	68 (58.62%)
	Absent	26 (22.41%)
**Rectovaginal nodule**	Dimension (cm)	1.58 ± 1.19
**Bladder nodule**	Dimension (cm)	2.33 ± 0.47
**rASRM classification**	Grade 3	34 (29.31%)
	Grade 4	82 (70.68%)
**ENZIAN Compartment A**	0	24 (20.68%)
	A1	11 (9.48%)
	A2	62 (53.44%
	A3	19 (16.37%)
**ENZIAN Compartment B**	0	25 (21.55%)
	B1	25 (21.55%)
	B2	62 (53.44%)
	B3	4 (3.44%)
**ENZIAN Compartment C**	0	68 (58.65%)
	C1	37 (31.89%)
	C2	11 (9.48%)
	C3	0
**ENZIAN FA**		5 (4.31%)
**ENZIAN FB**		4 (3.44%)

**Table 3 medicina-60-01534-t003:** Surgical procedures.

Procedure	Nr
Right ovarian cystectomy	69
Left ovarian cystectomy	92
Pouch of Douglas adhesiolysis	86
Complete excision of RV nodule	73
Incomplete excision of RV nodule	9
Uterosacral ligaments ablation	88
Vaginal wall resection for nodule	11
Bladder nodule excision	4

**Table 4 medicina-60-01534-t004:** Factors associated with QoSL.

		r (Pearson)	*p*
**History of Prior Surgery for Endometriosis**		0.1895	0.04
**Pouch of Douglas Status**	Unobliterated	−0.2023	0.02
	Partially Obliterated	−0.1226	0.18
	Completely Obliterated	0.2860	0.001
**Uterosacral Ligaments Status**	Not infiltrated	−0.2321	0.01
	Unilateral Infiltration	0.0378	0.68
	Bilateral Infiltration	0.1664	0.07
**rASRM grade 3**		−0.1116	0.23
**rASRM grade 4**		0.1116	0.23
**Adenomyosis**		−0.0318	0.73
**Pain Domain Severe Score**		0.2969	0.001

**Table 5 medicina-60-01534-t005:** Pain symptoms and QoSL before and after laparoscopy.

Mean Score	Prior to Surgery	2 Years after Surgery	*p*
**QoSL**	49.66	13.23	<0.0001
**Dyspareunia**	5.69	1.63	<0.0001
**Dysmenorrhea**	7.59	2.98	<0.0001
**Chronic pelvic pain**	3.68	1.07	<0.0001

**Table 6 medicina-60-01534-t006:** Pain symptoms and QoSL after complete and incomplete excision of rectovaginal nodule.

	Complete Excision of RV Endometriotic Nodule	Incomplete Excision of RV Endometriotic Nodule	*p*
**Median Sexual QoL Score**	12.39	25	0.02
**Dyspareunia VAS**	1.48	3.89	<0.0001
**Dysmenorhea VAS**	2.85	6.22	<0.0001
**Chronic pelvic pain VAS**	1.08	2.33	0.02

## Data Availability

Datasets analyzed or generated during the study are available on demand, contact the corresponding author.

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
