# Peer review of "Is Complete Excision Always Enough? A Quality of Sexual Life Assessment in Patients with Deep Endometriosis"

_medicina, 2024, doi:10.3390/medicina60091534_

Round 1

Reviewer 1 Report

Comments and Suggestions for Authors

Thank you for this manuscript:  
A few comments: 

Line 65: Please describe the location of the study. For example, a private hospital or teaching hospital etc  Line 66: Please provide details of the ethical insight provided by the hospital (Date and no.)  Line 77: Why has Virgo intacta excluded?  Line 82: Please include parity in the demographic data  Line 80: What was the dropout rate at 1 year? Lost to follow-up after 2 years at only 9 % is incredibly low.  Line 88: Why were 2-year responses more significant than 1 year?  LINE 100: Please describe the anaesthesia drug used and state that all patients the same drugs.  What were the surgeon's skills in performing these surgeries? How many surgeons were involved in this study?  I think it is easier to understand the flow of the study by creating a ‘Consort diagram’ Table 1” What does Education level Gymnasium mean?  Table 1: Please include parity  25% of the participants had previous surgery done What was the surgical difficulty level for these patients? Were there any intraoperative complications 

Fig 1 is best presented in a Table with P value  Table 4 legend is presented in point form. Any reason this is not standardized with the rest of the table/fig?

Author Response

Thank you for carefully reviewing the manuscript and for your constructive feedback; We feel that the manuscript has been significantly improved using your guidance.

All comments have been dealt with - changes are highlighted in yellow. 

Line 65: Please describe the location of the study. For example, a private hospital or teaching hospital etc 

This has been addressed.

Line 66: Please provide details of the ethical insight provided by the hospital (Date and no.) 

This has been addressed.

Line 77: Why has Virgo intacta excluded? 

This has been addressed – very difficult if not impossible to assess dyspareunia.

Line 82: Please include parity in the demographic data 

This has been addressed.

Line 80: What was the dropout rate at 1 year? Lost to follow-up after 2 years at only 9 % is incredibly low. 

This has been addressed – The first author has been in touch by phone with the patients and the questionnaire was sent online.

Line 88: Why were 2-year responses more significant than 1 year? 

This has been addressed.

LINE 100: Please describe the anaesthesia drug used and state that all patients the same drugs. 

This has been addressed.

What were the surgeon's skills in performing these surgeries?

How many surgeons were involved in this study? 

This has been addressed.

I think it is easier to understand the flow of the study by creating a ‘Consort diagram’ Table 1”

We feel that the diagram is superfluous given the fact this is not a randomized study and very few patients were excluded and/or lost.

What does Education level Gymnasium mean? 

This has been addressed – secondary studies.

Table 1: Please include parity  25% of the participants had previous surgery done

This has been addressed.

What was the surgical difficulty level for these patients?

Were there any intraoperative complications 

This has been addressed.

Fig 1 is best presented in a Table with P value  Table 4 legend is presented in point form. Any reason this is not standardized with the rest of the table/fig?

This issue has been addressed. Fig 1 displays the VAS evaluation for pain prior to surgery. Due to the discrepancy in pain medication and the effect of different medication on the individual patient and the huge bias this represents, no statistical relevance can be obtained, thus we decided to keep the visual (as it has been evaluated in VAS) representation in a figure with columns rather than a table. 

Reviewer 2 Report

Comments and Suggestions for Authors

Reviewer

This prospective experimental study is a very interesting idea. However some methodological and linguistic corrections have to be done. 

Comment 1: 

On line 34, you say that: The most common form of deep infiltrating endometriosis is rectovaginal endometriosis. However, the most common one is the involvement of uterosacral ligaments. See here: 

Non-invasive imaging techniques for diagnosis of pelvic deep

endometriosis and endometriosis classification systems:

an International Consensus Statement

G. CONDOUS1#, B.GERGES1,2# , I. THOMASSIN-NAGGARA3, C. BECKER4, C. TOMASSETTI5,6,

H. KRENTEL7 , B. J. VAN HERENDAEL8,9, M. MALZONI10, M. S. ABRAO11 , E. SARIDOGAN12,

J. KECKSTEIN13 , G. HUDELIST14 and Collaborators†

Comment 2:

Line 41. I would not say that I well known the correlation between the depth of the infiltration and the symptoms, as is still   discussion of a lot of studies. 

Comment 3:

Line 57. It not clear when you say that we do not have to focus only on symptoms but also on quality of life. Actually, the QoL of patients with endometriosis is based on symptoms, so this phrase is not clear. Secondary, you highlight a consensus of 2004, so 20 years ago to say that all visible endometriotic lesions have to be removed. This affirmation nowadays is not correct, especially when we operate women with infertility, one of the symptoms of endometriosis. 

Comment 4:

Line 66: Could you change prospective interventional study with prospective experimental study. 

Comment 5:

Line 69: You specify heterosexual patients, what about homosexual patients with endometriosis and deep dyspareunia? 

Comment 6:

Table 1. If is median the other parameter has to be quantiles. I think you are talking about media and standard deviation. 

Comment 7:

Figure 1: It is written pacients (number), could you correct that?

Comment 8: Correct corelated to correlated in line 22 and 230, 10 moth to 10 month in line 244. 

Comment 9: You mention ENZIAN classification in the materials but I do not see it in the results. 

Comment 10: There is one more 51. In the bibliography which is quit non updated, recent literature is required. 

Comment 11: 

It would be interesting to know if these women have already give birth and in which way of birth? (vaginal birth vs cesarean section vs operative birth). Furthermore, after surgery did they assume estroprogesteron or

Comments on the Quality of English Language

There are some phrases written in a not naive english language. 

Author Response

Thank you for carefully reviewing the manuscript and for your constructive feedback; We feel that the manuscript has been significantly improved using your guidance.

All comments have been dealt with - changes are highlighted in green. 

The manuscript underwent English proofreading.

Comment 1: 

On line 34, you say that: The most common form of deep infiltrating endometriosis is rectovaginal endometriosis. However, the most common one is the involvement of uterosacral ligaments. See here: 

Non-invasive imaging techniques for diagnosis of pelvic deep

endometriosis and endometriosis classification systems:

an International Consensus Statement

  1. CONDOUS1#, B.GERGES1,2# , I. THOMASSIN-NAGGARA3, C. BECKER4, C. TOMASSETTI5,6,
  2. KRENTEL7 , B. J. VAN HERENDAEL8,9, M. MALZONI10, M. S. ABRAO11 , E. SARIDOGAN12,
  3. KECKSTEIN13 , G. HUDELIST14 and Collaborators†

This has been revised and the suggested reading has been cited.

Comment 2:

Line 41. I would not say that I well known the correlation between the depth of the infiltration and the symptoms, as is still   discussion of a lot of studies. 

This has been revised and the statement rephrased.

Comment 3:

Line 57. It not clear when you say that we do not have to focus only on symptoms but also on quality of life. Actually, the QoL of patients with endometriosis is based on symptoms, so this phrase is not clear.

This statement has been revised.

Secondary, you highlight a consensus of 2004, so 20 years ago to say that all visible endometriotic lesions have to be removed. This affirmation nowadays is not correct, especially when we operate women with infertility, one of the symptoms of endometriosis. 

This has been revised – the statement has been revised and literature cited.  

Comment 4:

Line 66: Could you change prospective interventional study with prospective experimental study. 

This has been revised.

Comment 5:

Line 69: You specify heterosexual patients, what about homosexual patients with endometriosis and deep dyspareunia? 

This has been revised – difficult to assess dyspareunia in a standardized manner. Literature has been cited.

Comment 6:

Table 1. If is median the other parameter has to be quantiles. I think you are talking about media and standard deviation. 

This has been revised

Comment 7:

Figure 1: It is written pacients (number), could you correct that?

This has been revised

Comment 8: Correct corelated to correlated in line 22 and 230, 10 moth to 10 month in line 244. 

This has been revised

Comment 9: You mention ENZIAN classification in the materials but I do not see it in the results. 

This has been revised – see table 2.

Comment 10: There is one more 51. In the bibliography which is quit non updated, recent literature is required. 

This has been revised – references have been updated; 51 was the page of the last reference

Comment 11: 

It would be interesting to know if these women have already give birth and in which way of birth? (vaginal birth vs cesarean section vs operative birth). Furthermore, after surgery did they assume estroprogesteron or

This has been revised

Round 2

Reviewer 2 Report

Comments and Suggestions for Authors

In my opinion the changes, i previously have suggest, are clearly done.

The topic of endometriosis remain enigmatic and very interesting. Future research has to be done to evaluate better the QoF in women suffering from endometriosis.